# Analysis of Biochemical and Antimicrobial Properties of Bioactive Molecules of *Argemone mexicana*

**DOI:** 10.3390/molecules28114428

**Published:** 2023-05-30

**Authors:** Jyotsna Jaiswal, Nikhat J. Siddiqi, Sabiha Fatima, Manal Abudawood, Sooad K. AlDaihan, Mona G. Alharbi, Maria de Lourdes Pereira, Preeti Sharma, Bechan Sharma

**Affiliations:** 1Department of Biochemistry, Faculty of Science, University of Allahabad, Allahabad 211002, India; 2Department of Biochemistry, College of Science, King Saud University, Riyadh 11495, Saudi Arabia; 3Department of Clinical Laboratory Sciences, College of Applied Medical Sciences, King Saud University, Riyadh 11433, Saudi Arabia; 4Department of Medical Sciences, CICECO—Aveiro Institute of Materials, University of Aveiro, 3810-193 Aveiro, Portugal; 5Department of Biotechnology, Veer Narmad South Gujarat University, Udhna Magdalla Road, Surat 395007, India; preetisharma@vnsgu.ac.in

**Keywords:** *Argemone mexicana*, antioxidant, phytochemicals, antimicrobial activity, anti-HIV-1 reverse transcriptase

## Abstract

This study identified phytochemicals in *Argemone mexicana (A. mexicana)* extracts that are responsible for its medicinal properties, and the best solvent for their extraction. The extracts of the stem, leaves, flowers, and fruits of *A. mexicana* were prepared at low (corresponding to room temperature) and high temperatures (corresponding to the boiling points) in various solvents, viz., hexane, ethyl acetate, methanol, and H_2_O. The UV-visible absorption spectra of various phytoconstituents in the isolated extracts were determined through spectrophotometry. Qualitative tests for the screening of phytoconstituents in the extracts were performed to identify various phytochemicals. We identified the presence of terpenoids, alkaloids, cardiac glycosides, and carbohydrates in the plant extracts. The antioxidant and anti-human immunodeficiency virus type 1 reverse transcriptase (anti-HIV-1RT) potential, as well as the antibacterial activity of various *A. mexicana* extracts were determined. These extracts showed strong antioxidant activities. The extracts exhibited antimicrobial activities against *Salmonella typhi*, *Staphylococcus epidermis*, *Citrobacter*, *Neisseria gonorrhoeae*, and *Shigella flexineri*. These extracts significantly inhibited HIV-1 reverse transcriptase activity. The aqueous leaf extract prepared at a temperature equivalent to the boiling point, i.e., 100 °C, was identified to be the most active against pathogenic bacteria and HIV-1 RT.

## 1. Introduction

*Argemone mexicana* L. *(A. mexicana)* (Papaveraceae), also known as the Mexican prickly poppy or Mexican poppy, grows in the tropical and subtropical regions globally. *Argemone mexicana* L. is mainly found in Mexico but it is now widely distributed across many parts of the world, including India, Bangladesh, the United States and Ethiopia [1,2]. *A. mexicana*, a globally used medicinal plant, serves as a source of many alternative medicines. The herbs/shrubs and many trees present in the wild state possess a huge number of novel phytochemicals of medicinal significance. Studies on the wild species of certain weeds have been considered of great importance for the treatment of various diseases [3].

The yellow exudate of *A. mexicana* has been used to treat dropsy, jaundice, scabies, and skin diseases [4,5]. The flowers, leaves, and seeds of this plant have been used to treat diverse diseases including dysentery, ulcers, cough, and hypertension [5,6,7,8,9]. *A. mexicana* also exhibits hepatoprotective, anticancer, anti-HIV, antiproliferative, anti-inflammatory, antibacterial, antidiabetic, antifertility, antiallergic, nematocidal, and antioxidant activities [10,11]

The leaves and stems of *A. mexicana* are employed to treat malaria and dropsy. They possess anti-analgesic, antispasmodic, antiparasitic, and narcotic properties with antifungal, hepatoprotective, larvicidal, and chemosterilant activities [12,13]. The leaf extract has been used as a disinfectant, whereas the seed extract has been employed for the treatment of leprosy, warts, skin diseases, and insect bites. These activities are attributed to secondary metabolitesand protein hydrolyzing substances [1,14].

The extracts of different parts of *A. mexicana* are used in different forms. For example, the root paste is used to treat insect and scorpion bites, etc., in the form of ointment for external use for the treatment of wound healing. For constipation and bloating, the root powder is used orally at a dose of 1–2 g/day. The fruit extracts of *A. mexicana* have been used as intra-peritoneal injections in anticancer studies in mice [15]. The latex from *A. mexicana* is useful to treat conjunctivitis, while the oil from the seeds is used to treat asthma, dysentery, and ulcers, etc. [16].

The secondary metabolites of *A. mexicana* such as flavonoids, polyphenols, phenols, alkaloids, saponins, and tannins, have been shown to be responsible for exhibiting medicinal properties. These secondary metabolites isolated from different parts of *A. mexicana* contain antimicrobial activities against different species of fungal, bacterial, and viral pathogens [11,17]. 

The biomedical properties of these phytochemicals isolated in different solvents at high and low temperatures have not been properly studied. In view of this background and the traditional uses of *A. mexicana* as a medicinal plant, the aim of this study was to optimize the best solvent and temperature for the extraction of secondary metabolites from different parts, namely the leaves, stem, fruits, and flowers of *A. mexicana*. We have attempted to characterize the chemical ingredients present in the different extracts of the plant through chemical analysis, monitoring their retardation factor (Rf) values employing thin layer chromatography (TLC), and determining the differential absorption of light at varying UV and visible ranges. Further, it has been endeavored to assess their antimicrobial, and antiviral (especially anti-HIV reverse transcriptase) activities.

## 2. Results

### 2.1. Spectrophotometric Analysis of Phytochemicals from A. mexicana Extracts at Room Temperature

We noted major absorption peaks at 276, 370, 411, and 669 nm, indicating that flavonoids were the major compounds in the *A. mexicana* leaf extract prepared at room temperature in hexane, ethyl acetate, methanol and aqueous medium. Pheophytin A was the second most abundant chemical compound present in the leaf extract (peaks observed at 370, 411, and 669 nm). However, these peaks were not noted in the aqueous extract. Unsaturated carbonyl compounds were found in only the aqueous leaf extract at 198 nm and at a lower concentration of the extract (250 µg/mL).

The stem extracts prepared at room temperature in hexane, ethyl acetate, methanol, and aqueous medium revealed major peaks at 332, 340, 410, and 669 nm, indicating the presence of flavonoids. Pheophytin A was present only in a few sections of the extract at 371, 402, and 667 nm. The unsaturated carbonyl compounds were found in only the aqueous stem extract at 340, 415, and 669 nm and at a lower concentration (250 µg/mL). The fruit extract prepared at room temperature in hexane, ethyl acetate, methanol, and aqueous medium revealed peaks at 206, 232, 333, 301, 410, and 667 nm, indicating the occurrence of pheophytin A and flavonoids. Unsaturated carbonyl compounds were detected at 206 and 221 nm. (Appendix A).

### 2.2. Spectrophotometric Analysis of Phytochemicals from A. mexicana Extracts at a High Temperature

We noted the peaks in the visible range (190 to 700 nm), which indicated the occurrence of phytochemicals. In the leaf extract produced at a high temperature in hexane, ethyl acetate, methanol, and aqueous medium, the most abundant compounds were flavonoids, with absorption peaks observed at 231, 236, 405, 666, and 668 nm. Pheophytin A (absorbance peaks at 237, 376, and 669 nm) and the unsaturated carbonyl compounds (peak at 205 nm) were also observed.

The stem and fruit prepared at a high temperature in hexane, ethyl acetate, methanol, and aqueous medium extracts contained flavonoids, with absorption peaks noted at 227, 350, 380, 666, and 669 nm. Pheophytin A was the second most abundant phytochemical in the stem extract across all the solvents; the peak was noted at 422 nm. Pheophytin A was not present in the methanolic extract of the fruit, and the extracts of stem and fruit prepared in water. The unsaturated carbonyl compounds were found in the aqueous stem extract at 208 nm only. In the flower extract, flavonoids and pheophytin A were identified (239, 351, and 669 nm) in the methanolic extract, and unsaturated carbonyl compounds at 210 nm at lower solvent concentrations (Appendix A).

### 2.3. Phytochemical Analysis of Phytochemicals from A. mexicana Extracts at Room Temperature as Detected by TLC

To confirm the results obtained from spectral analysis, TLC of the fractions which showed the presence of optimal concentrations of phytochemicals was carried out. These results are presented in Figure 1. The ethyl acetate fruit extract of *A. mexicana* showed the presence of alkaloids, flavonoids, primary and secondary amines. Other phytochemicals included antioxidants, carbohydrates, saponins, glycosides, essential oil, phenols, mycotoxin, and terpenes.

### 2.4. Phytochemical Analysis of A. mexicana Extracts Prepared at the Boiling Points of Solvents (High Temperature) Using TLC

The results are shown in Figure 2. The extract of *A. mexicana* leaves produced in water at 100 °C indicated the occurrence of flavonoids and the primary/secondary amines. Antioxidants, glycosides, steroids, carbohydrates, prostaglandins, saponins, essential oil, phenols, and terpenoids were also detected.

The methanolic fruit extract showed the presence of prostaglandins, antioxidants, carbohydrates, steroids, mycotoxin, phenols, saponins, terpenes, glycosides, and essential oil (Figure 3).

### 2.5. Chemical Characterization of A. mexicana Extracts

#### 2.5.1. Qualitative Detection of the Bioactive Compounds Present in the Extracts Produced at Room Temperature

Table 1 lists the various phytochemicals identified in different *A. mexicana* extracts prepared in diverse solvents at room temperature and high temperature.

The leaf extract prepared in these solvents contained terpenoids. The ethyl acetate extract contained cardiac glycosides, phenols, and traces of flavonoids. The aqueous and methanolic leaf extracts contained carbohydrates, phenols, quinones, and traces of flavonoids.

The stem extract of *A. mexicana* contained only terpenoids, flavonoids, quinones, and small amounts of steroids. Carbohydrates, cardiac glycosides, flavonoids, phenols, quinones, steroids, and terpenoids were detected in the ethyl acetate extract made at a low temperature. The methanolic extract produced at room temperature contained high amounts of alkaloids, carbohydrates, flavonoids, phenols, quinones, saponins, steroids, and terpenoids. The aqueous extract prepared under this condition contained only carbohydrates, phenols, and quinones.

The fruit extract obtained in hexane at a low temperature contained cardiac glycosides, quinones, terpenoids, and phenols. The ethyl acetate extract contained cardiac glycosides, steroids and terpenoids, phenols, and quinones, with traces of carbohydrates. The methanolic fruit extract contained alkaloids, carbohydrates, cardiac glycosides, flavonoids, quinones, steroids, terpenoids, and traces of phenol, whereas the aqueous extract contained carbohydrates, flavonoids, terpenoids, tannins, phenol, and quinones.

The flower extract obtained in various solvents at a low temperature did not have significant phytochemicals (unpublished data).

#### 2.5.2. Qualitative Analysis of the Phytochemicals Present in Different *A. mexicana* Extracts Prepared at a High Temperature

The leaf extract in hexane obtained at a high temperature contained terpenoids, quinones, and alkaloids, with carbohydrates and cardiac glycosides in small amounts. The ethyl acetate leaf extract contained cardiac glycosides, terpenoids and small amounts of phenols and flavonoids. The leaves extracted in methanol at a high temperature contained high amounts of alkaloids, carbohydrates, phenols, terpenoids, and tannins. Similar phytochemicals were found in the aqueous leaf extract made at 100 °C, except for small amounts of quinones.

The stem extracted in hexane at a high temperature contained carbohydrates, cardiac glycosides, quinones, and terpenoids in significant amounts, followed by phenols and traces of flavonoids. The ethyl acetate extract contained cardiac glycosides, terpenoids, quinones, phenols, and small amounts of flavonoids. Methanol and aqueous stem extracts prepared at a high temperature contained significant amounts of alkaloids, carbohydrates, phenols, terpenoids, and tannins. The aqueous extract also contained quinones.

The fruit extract prepared in hexane contained cardiac glycosides, quinones, steroids, terpenoids, and small amounts of flavonoids and phenols. The ethyl acetate extract contained cardiac glycosides, steroids, and terpenoids, as well as small amounts of carbohydrates, phenols, and quinones. Methanolic and aqueous fruit extracts prepared at a high temperature contained significant amounts of carbohydrates, phenols, quinones, and terpenoids. Steroids were present in methanolic extracts, but absent in aqueous extracts. The aqueous extracts, but not methanolic extracts, contained alkaloids and tannins.

Only the flower extracts prepared using methanol and water had detectable phytochemicals. The methanolic flower extract obtained at a high temperature contained significant amounts of carbohydrates, phenols, quinones, alkaloids, cardiac glycosides, terpenoids, and traces of flavonoids. The aqueous extract contained carbohydrates, phenols, quinones, tannins, and terpenoids, as well as small amounts of alkaloids and flavonoids (Table 1).

#### 2.5.3. Free-Radical Neutralizing Ability of *A. mexicana* Extracts

The leaf extract revealed the highest free-radical scavenging activity in water, followed by methanol and ethyl acetate at room temperature and the temperature equivalent to their boiling points. The free-radical neutralizing ability of stem extracts in various solvents were as follows: methanol > aqueous > ethyl acetate in extracts prepared at low temperatures. The free-radical quenching activity of the stem extracts obtained at high temperatures was in the following order: ethyl acetate > water > methanol. The fruit extract made at room temperature exhibited similar activity in methanol and water. The fruit extract made at a temperature equivalent to their boiling point revealed similar activity in water and methanol, which was less than that in ethyl acetate. The flower extract displayed free-radical scavenging activity only when made at a temperature equivalent to the boiling points in methanol and water. The IC_50_ values are listed in Table 2.

### 2.6. Effect of A. mexicana Extract against the Activity of HIV-1 RT

The leaf extract prepared in distilled water at a high temperature exhibited the maximum inhibitory effect against HIV-1 RT, with an IC_50_ value of 0.044 mg/mL, followed by ethyl acetate and methanolic extracts. Only the leaf extract prepared in the ethyl acetate at room temperature displayed HIV-1 RT activity (Table 3). Similarly, only the ethyl acetate fruit extract prepared at high and low temperatures exhibited HIV-1 RT activity. No activity of anti-HIV-RT was detected in other extracts.

### 2.7. Antimicrobial Activity of Various A. mexicana Extracts

The antimicrobial effects of different extracts prepared at room temperature against different bacterial strains are listed in Table 4. Ampicillin (5 mg/mL) was used as a standard. The ethyl acetate and methanolic leaf extracts prepared at a low temperature were effective against *S. typhi*, *Citrobacter*, *Shigella flexineri*, and *S. epidermis. Neisseria gonorrhoeae* was inhibited only by the ethyl acetate leaf extract synthesized at room temperature. The ethyl acetate and methanolic fruit extracts prepared at room temperature inhibited *Salmonela typhi*, *Neisseria gonorrhoeae*, and *S. flexineri. S. epidermis* was inhibited only by the methanolic fruit extract.

Table 5 presents the antimicrobial effects of the plant extracts synthesized at the temperature equivalent to the boiling points of the solvents. Methanol and aqueous leaf extracts prepared at high temperatures inhibited *Gonococci* and *Citrobacter. Salmonella typhi* was inhibited by aqueous and ethyl acetate extracts, and *Staphylococcus epidermis* was inhibited only by the ethyl acetate leaf extract. Methanolic and ethyl acetate fruit extracts prepared at a high temperature inhibited *Salmonella typhi* and *Citrobacter.* Only the methanolic fruit extract inhibited *Gonococci* and *S. flexineri*, whereas *S. epidermis* was inhibited only by the ethyl acetate fruit extract.

## 3. Discussion

*A. mexicana* belongs to the Papaveraceae family, which possesses antibacterial [18,19,20], antifungal [21,22], antiviral [23,24], antioxidant [25,26], and cytotoxic/anticancer [27] properties. The emergence of drug-resistant microbial strains [28] has necessitated the search for antimicrobial compounds from natural sources, such as plants and marine microorganisms. Many studies have determined the antimicrobial effects and secondary metabolites of *A. mexicana* [29,30]. Recently, A study by Orozco-Nunnelly et al., 2021 [28] has shown that the methanolic extract of outer roots and leaves displayed strong antimicrobial activities, especially against Gram-positive bacteria. In this study, different extracts of *A. mexicana* have been shown tocause an inhibition in the growth of both Gram +ve and Gram −ve bacteria. The results from the present study revealed that the temperatures corresponding to the solvents’ boiling points resulted in a better yield of phytochemicals. Terpenoids were the most abundant, followed by phenols, flavonoids, carbohydrates, and cardiac glycosides, as also reported by [29]. The leaf and the stem extracts exhibited higher free-radical scavenging activity than the fruit and flower extracts, which may be attributed to phenolic compounds [31]. The leaf extract possessed antioxidant activity, as also shown by Perumal et al. (2010) [32].

HIV uses reverse transcriptase (RT) to convert its RNA to cDNA. Ishizuka et al., 2020 [23] have shown that RT exhibits RNA- and DNA-dependent DNA polymerase activities and RNase H activity. In the present study, the aqueous extract of leaves prepared at a high temperature indicated the maximum inhibitory activity against HIV-1 RT, much like any other nonnucleoside reverse transcriptase inhibitor (NNRTI), as demonstrated by Sanna et al., 2018 [33]. The fruit extracts prepared at a high temperature also exhibited anti-HIV-RT activity, which may be attributed to the presence of flavonoids and alkaloids in these extracts [34]. Nuciferine, an alkaloid in the roots of *A. mexicana*, has been reported to be responsible for its anti-HIV-1RT activity [35]. The methanolic extracts of the entire *A. mexicana* contains benzo[c]phenanthridine alkaloid and (±)-6-acetonyl dihydrochelerythrine, which exhibited anti-HIV activity [35]. The anti-HIV-RT property was found to be mainly associated with the aqueous extract of the leaves at a high temperature. The results from earlier studies in collaboration with other works have shown anti-HIV-RT activity by some organic compounds present in the leaf extracts, displaying an inhibition of HIV-1 DNA polymerase function [23]. Various plant extracts exhibited antimicrobial activities against different test organisms, namely *S. typhi*, *Gonococci*, *Citrobacter*, *S. flexineri*, and *S. epidermis.* Among the extracts prepared at room temperatures, the methanolic extract revealed the best antimicrobial activity. On the other hand, among the extracts prepared at high temperatures, the aqueous extract revealed the maximum antimicrobial activity. The antimicrobial activity may be attributed to phenols, terpenoids, and alkaloids [36].

The methanolic extracts of the outer root of *A*. *mexicana* possessed maximum activity against Gram+ve bacteria, and minimum activity against the Gram−ve bacteria and fungi (Orozco-Nunnelly et al., 2021) [28]. The antimicrobial activity of the leaf extract can be attributed to the presence of berberine, an alkaloid [28]. Berberine exerts its effect by damaging the cell membrane and impeding the synthesis of proteins and DNA [37]. Berberin chloride, in combination with various anti-staphylococcal drugs as a reference of CoNS strains, varied greatly across different bacterial stains and drugsused [38]. Rahman et al., 2011 [19] reported that the leaf extract of *A. mexicana* inhibited *Salmonella sp*, which was also noted in this study.

## 4. Materials and Methods

### 4.1. Reagents and Chemicals

The reagents Luria–Bertani broth (Miller, Appleton, WI, USA) and Mueller–Hinton agar were procured from SRL Pvt. Ltd. (Mumbai, India), Himedia, whereas the ampicillin sodium injection was obtained from Saralife (New Delhi, India). Silica-gel-coated TLC plates, Wagner’s reagent, and Molisch reagent were procured from Merck, Darmstadt, Germany. All other reagents used were of analytical grade.

### 4.2. Plant Material Collection and Extract Preparation

*A. mexicana* plants were collected from Prayagraj, India, and nearby areas. The leaves, stems, flowers, and fruits were collected. The collected plant materials were washed under running tap water and then with double-distilled water. The plant parts were then dried in shade and powdered. The powder was dissolved in the ethyl acetate, hexane, water, and methanol at a low temperature (room temperature) and high temperature (temperatures equivalent to the boiling points of the solvents such as 77.1 °C, 68 °C, 100 °C, and 64.7 °C, respectively). The extracts at a high temperature were prepared using the Soxhlet apparatus. These extracts were dried at room temperature and used by dissolving them into the desired volume of the solvents for using them at different concentrations for carrying out different specific experiments, or stored at −20 °C until further use.

### 4.3. Analysis of A. mexicana Extract Spectrophotometrically

The spectrophotometric analysis of the extracts resulted in different spectral patterns corresponding to the specific absorption profiles under UV and visible wavelengths of various components present in the isolated extracts determined through spectrophotometry. The UV-visible spectra of different concentrations of extracts were placed in a quartz cuvette and recorded from 190 to 750 nm by using the double-beam UV-visible spectrophotometer (model: Scientific Spectroscan UV2700, Thermo Fisher Scientific India Pvt. Ltd., Pune, India) [39,40]. The analysis of the absorption peaks was carried out and compared with the peaks of specific bioactive compounds.

### 4.4. Thin Layer Chromatography (TLC)

The TLC plates coated with the silica gel (Merck, Darmstadt, Germany) were employed for the analysis of different phytochemicals present in the various extracts of *A. mexicana.* The fractions of which the UV-visible absorption spectra showed the presence of optimal concentrations of phytochemicals were subject to thin layer chromatography to confirm the results.

### 4.5. Assay of Phytochemicals

Phytoconstituents including alkaloids, carbohydrates, cardiac glycosides, flavonoids, phenols, steroids, and saponins present in the extracts were qualitatively evaluated using previously described standard procedures [41,42]. 

### 4.6. 2,2-Diphenyl-1-picrylhydrazyl Assay

The free radical quenching potential of various extracts was determined using the 2, 2-diphenyl-1-picrylhydrazyl assay (DPPH), as described previously [43], with some modifications. Of the 10 mM DPPH solution, 1 mL was added to different extract concentrations (25, 50, 100, 200, 300, and 500 µg/mL). After 30 min incubation at room temperature, the absorbance was determined at 517 nm. The ascorbic acid (1%) was used as a standard. The free radical scavenging capacity of the extracts was calculated as shown below:Scavenging activity of DPPH (%) = [(Absorption by control − Absorption by the sample)/(Absorption by control) × 100]

### 4.7. Examination of Different Plant Extracts for Their Anti-HIV-1RT Activity

The anti-HIV-1RT activity of different plant extracts was determined by employing RT colorimetric assay kits (Roche, NY, USA), in accordance with the manufacturer’s instruction.

### 4.8. Antimicrobial Activity

#### 4.8.1. Test Microorganisms

Clinically pathogenic bacteria, namely *Neisseria gonorrhoeae*, *Salmonella typhi*, *Citrobacter*, *Staphylococcus epidermidis*, and *Shigella flexineri*, were used. These organisms were obtained from the Medical Sciences Institute of BHU, Varanasi, India.

#### 4.8.2. Antibacterial Activity

The agar-well diffusion method [44] was used to examine antibacterial susceptibility. Sterilized discs (6 mm) soaked in different extract concentrations (100, 200, 300, and 500 μg in 5 μL volume each) of extracts were placed on solid media with sterile forceps. Water and ampicillin (5 μg/mL) were employed as the negative and positive controls, respectively. The petri plates were placed for 24 h at 37 °C. The diameter of the zone of inhibition (mm) after 24 h incubation was used as a measure of antimicrobial activity.

### 4.9. Statistical Analysis

We performed all experiments in triplicate. Data are presented as the mean ± SE of the three replicates. All the data were statistically analyzed using (MS Excel)—Microsoft 365. (Graph pad Prism)—GraphPad Software, Inc., San Diego, CA, USA.

## 5. Conclusions

In the present study, the leaf extract of *A. mexicana* prepared in an aqueous medium at high temperatures was found to be the most active against HIV 1-RT and the tested microorganisms. Water at a high temperature was the best solvent to extract the pharmacological activities of bioactive molecules from *A. mexicana.* The pharmacological properties of *A. mexicana* may be attributable to various compounds, which may exert their actions through specific and nonspecific mechanisms. Therefore, active chemical components should be identified to elucidate their mechanisms of actions for the scientific utilization of the plant to cure various diseases.

## Figures and Tables

**Figure 1 molecules-28-04428-f001:**
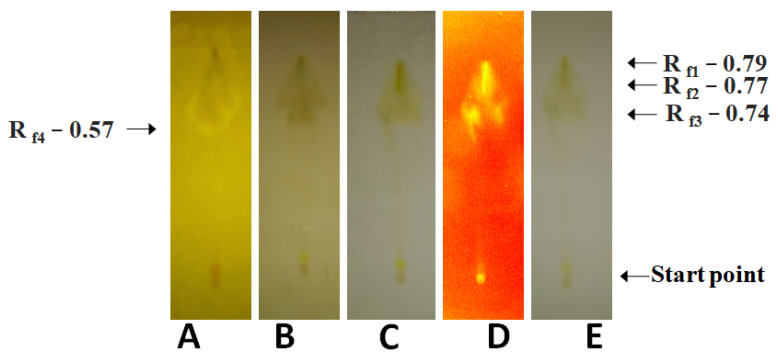
Phytochemical analysis of the *A. mexicana* fruits prepared in ethyl acetate at room temperature. (**A**): Dragendorff’s reagent treatment detecting alkaloids, and primary/secondary amines, (**B**): anisaldehyde–sulfuric acid reagent treatment detecting various natural products, (**C**): Mayer’s reagent treatment detecting alkaloids, (**D**): UV light treatment detecting photoactive compounds, and (**E**): visible light treatment detecting photoactive compounds.

**Figure 2 molecules-28-04428-f002:**
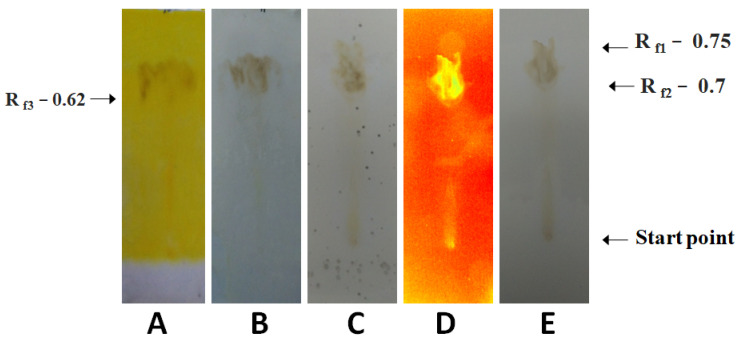
Analysis of phytochemicals present in the *A*. *mexicana* leaf extract prepared in water at a high temperature (100 °C). (**A**): Dragendorff’s reagent treatment detecting alkaloids, and primary/secondary amines, (**B**): anisaldehyde–sulfuric acid reagent treatment detecting various natural products, (**C**): Mayer’s reagent treatment detecting alkaloids, (**D**): UV light treatment detecting photoactive compounds, and (**E**): visible light treatment detecting photoactive compounds.

**Figure 3 molecules-28-04428-f003:**
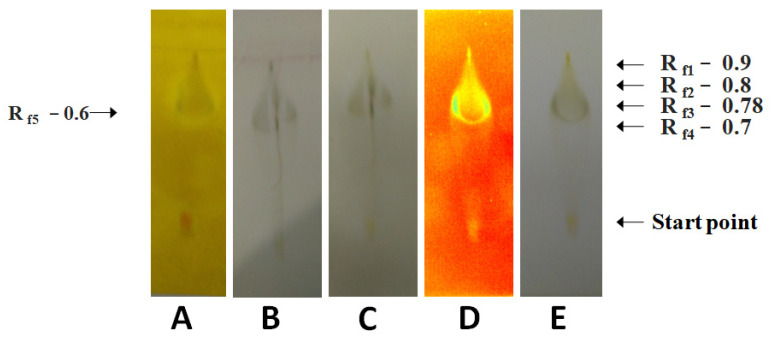
The analysis of the phytochemicals occurring in the extract of *A*. *mexicana* fruits prepared in the methanol at its boiling point (high temperature). (**A**): Dragendorff’s reagent treatment detecting alkaloids, and primary/secondary amines, (**B**): anisaldehyde–sulfuric acid reagent treatment detecting various natural products, (**C**): Mayer’s reagent treatment detecting alkaloids, (**D**): UV light treatment detecting photoactive compounds, and (**E**): visible light treatment detecting photoactive compounds.

**Table 1 molecules-28-04428-t001:** Qualitative estimation of the bioactive compounds occurring in the extracts of *Argemone mexicana* prepared at room and high temperatures.

Room Temperature											
Plant Name and Part	Solvent	Phytochemicals and Their Intensities									
		Alkaloids	Carbohydrates	Cardiac Glycosides	Flavonoids	Phenols	Quinones	Saponins	Steroids	Tannins	Terpenoids
** *Argemone mexicana* ** **Leaf**	Hexane	ND	ND	ND	ND	ND	++	ND	ND	ND	++
	Ethyl acetate	ND	ND	++++	+	++	ND	ND	ND	ND	+++
	Methanol	ND	++++	ND	+	+++	+++	ND	+	ND	+
	Aqueous	ND	++++	ND	+	+++	+	ND	ND	ND	++
** *Argemone mexicana* ** **stem**	Hexane	ND	ND	ND	+	ND	+	ND	++	ND	+
	Ethyl acetate	ND	++	++	+	++	++	ND	++	ND	++
	Methanol	++++	++++	ND	+++	++	+++	++	++++	ND	+++
	Aqueous	ND	++++	ND	ND	++	++	ND	ND	ND	ND
** *Argemone mexicana* ** **Fruit**	Hexane	ND	ND	+++	ND	++	+++	ND	ND	ND	+++
	Ethyl acetate	ND	+	++++	ND	++	++	ND	++++	ND	+++
	Methanol	+++	++++	++++	++	+	++++	ND	+++	ND	+++
	Aqueous	ND	++++	ND	+++	++	+	ND	ND	++	++++
**High temperature (solvent boiling point)**											
** *Argemone mexicana* ** **leaf**	Hexane	++	+	+	ND	ND	++	ND	ND	ND	++
	Ethyl acetate	ND	ND	+++	+	++	ND	ND	ND	ND	+++
	Methanol	++++	+++	ND	ND	+++	ND	ND	ND	++	++++
	Aqueous	++++	+++	ND	ND	+++	++	ND	ND	ND	+++
** *Argemone mexicana* ** **Stem**	Hexane	ND	+++	+++	+	++	+++	ND	ND	ND	+++
	Ethyl acetate	+	++	++++	+++	+++++	++++	ND	ND	ND	+++
	Methanol	++++	+++	ND	+	+++	ND	ND	ND	++	++++
	Aqueous	++++	+++	ND	ND	+++	+++	ND	ND	+++	++
** *Argemone mexicana* ** **Fruit**	Hexane	ND	ND	+++	++	++	+++	ND	+++	ND	+++
	Ethyl acetate	ND	++	+++	ND	++	++	ND	+++	ND	+++
	Methanol	ND	+++	++++	+	+	+++	ND	+++	ND	+++
	Aqueous	+++	++++	ND	+	+++	+++	ND	ND	++++	+++
** *Argemone mexicana* ** **Flower**	Methanol	++	+++	++	+	+++	+++	ND	ND	ND	++
	Aqueous	++	++++	ND	++	+++	+++	ND	ND	+++++	++++

+ denotes the intensity of the chemical ingredients present in the extracts. Increase in the number of (+) indicates an increase in the intensity. ND—not detected.

**Table 2 molecules-28-04428-t002:** Free-radical neutralizing ability of *Argemone mexicana* extracts prepared at room temperature and high temperature.

*Argemone mexicana*	Solvent	IC_50_ (µg/mL)	
		Room Temperature	High Temperature
**Leaf**	Aqueous	160	85
	Hexane	ND	ND
	Ethyl acetate	290	315
	Methanol	160	165
**Stem**	Aqueous	165	190
	Hexane	ND	ND
	Ethyl acetate	210	185
	Methanol	135	270
**Fruit**	Aqueous	240	235
	Hexane	ND	ND
	Ethyl acetate	ND	125
	Methanol	225	230
**Flower**	Aqueous	ND	145
	Methanol	ND	165

IC_50_—plant extract concentration to neutralize the free radicals by 50%. ND—not detected. The high temperature corresponds to the respective solvent’s boiling point.

**Table 3 molecules-28-04428-t003:** Effect of *Argemone mexicana* extracts prepared at a high temperature and room temperature on HIV-1 RT activity.

Plant Name	Part of Plant	Solvent	Extract Concentration (mg/mL)	Activity Lost (% Inhibition)	IC_50_ (mg/mL)
*Argemone mexicana*	Leaf	Hexane (HT)	0.083	5	ND
			0.166	10	
			0.332	11	
			0.664	14	
		Ethyl acetate (HT)	0.083	16	0.36
			0.166	24	
			0.332	48	
			0.664	78	
		Methanol (HT)	0.083	12	0.35
			0.166	25	
			0.332	47	
			0.664	71	
		Aqueous (HT)	0.415	47	0.044
			0.083	79	
			0.166	97	
			0.332	100	
			0.664	100	
		Hexane (RT)	0.083	ND	ND
			0.166	11	
			0.332	13	
			0.664	16	
		Ethyl acetate (RT)	0.083	11	0.39
			0.166	24	
			0.332	46	
			0.664	67	
		Methanol (RT)	0.083	ND	ND
			0.166	13	
			0.332	33	
			0.664	40	
		Aqueous (RT)	0.083	ND	ND
			0.166	27	
			0.332	33	
			0.664	24	
	Fruit	Hexane (HT)	0.083	ND	ND
			0.166	ND	
			0.332	16	
			0.664	36	
		Ethyl acetate (HT)	0.083	44	0.15
			0.166	51	
			0.332	47	
			0.664	55	
		Methanol (HT)	0.083	ND	ND
			0.166	ND	
			0.332	22	
			0.664	46	
		Aqueous (HT)	0.083	ND	ND
			0.166	ND	
			0.332	33	
			0.664	41	
		Hexane (RT)	0.083	ND	ND
			0.166	ND	
			0.332	11	
			0.664	26	
		Ethyl acetate (RT)	0.083	25	0.26
			0.166	29	
			0.332	71	
			0.664	84	
		Methanol (RT)	0.083	ND	ND
			0.166	ND	
			0.332	21	
			0.664	34	
		Aqueous (RT)	0.083	ND	ND
			0.166	11	
			0.332	24	
			0.664	30	

ND—not detected, RT—room temperature, HT—high temperature.

**Table 4 molecules-28-04428-t004:** *Argemone mexicana* extracts prepared at room temperature, showing antibacterial properties against various pathogenic bacterial strains.

Plant Part	Solvent	Bacterial Strain	Diameter of the Zone of Inhibition (mm)						
			Extracts (mg/mL)						
			0	50	100	200	300	500	Ampicillin (5 µg/mL)
Leaf	Ethyl acetate	*Salmonella typhi*	0		8	10	11	-	13
	Methanol		0		7	11	12	-	14
	Aqueous		0		-	-	-	-	-
Fruit	Ethyl acetate		0	-	7	10	13		15
	Methanol		0	-	7	9	10		14
	Aqueous		0	-	-	-	-	-	15
Leaf	Aqueous	*Neisseria gonorrhoeae*			-	-	-	-	15
	Ethyl acetate		0		7	10	13	-	15
	Methanol				-	-	-	-	15
Fruit	Aqueous		0	-	-	-	-	-	15
	Ethyl acetate		0	-	7	8	8	-	13
	Methanol		0	6	10	7	8	-	14
Leaf	Ethyl acetate	*Citrobacter*	0	-	12	12	14	-	14
	Methanol		0	6	11	11	12	13	14
	Aqueous			-	-	-	-	-	14
Fruit	Ethyl acetate		0	-	-	-	-	-	14
	Methanol		0	-	-	-	8	-	13
	Aqueous		0	-	-	-	-	-	15
Leaf	Ethyl acetate	*Shigella f* *lexineri*	0	-	7	-	8	8	12
	Methanol		0	-	7	10	12	12	14
	Aqueous			-	-	-	-	-	15
Fruit	Ethyl acetate		0	7	-	7	9	-	13
	Methanol		0	-	7	8	-	-	13
	Aqueous		0	-	-	-	-	-	14
Leaf	Ethyl acetate	*Staphylococcus* *epidermis*	0	-	7	-	9	9	13
	Methanol		0	-	10	11	13	13	15
	Aqueous			-	-	-	-	-	14
Fruit	Ethyl acetate		0	-	-	-	-	-	11
	Methanol		0	7	10	-	-	-	13
	Aqueous		0	-	-	-	-	-	15

**Table 5 molecules-28-04428-t005:** Effects of the extracts of leaves and fruits of *Argemone mexicana* prepared at a high temperature on various pathogenic bacterial strains.

Plant name and Part	Solvent	Bacterial Strain	Zone of Inhibition (mm)						
			Extracts (mg/mL)						
			0	50	100	200	300	500	Ampicillin (5 µg/mL)
*A. mexicana leaf*	Ethyl acetate	*Salmonella typhi*	0	8	8	10	8		16
	Methanol		-	-	-	-	-		16
	Aqueous		0	-	12	13	15		16
*A. mexicana fruit*	Ethyl acetate		0	6	7	7	8		16
	Methanol		0	-	6	7	11		12
	Aqueous		-	-	-	-	-		12
*A. mexicana leaf*	Ethyl acetate	*Gonococci*	-	-	-	-	-	-	
	Methanol		0	6	7	10	12		14
	Aqueous		0	-	11	13	14		14
*A. mexicana fruit*	Ethyl acetate		0	-	-	-	-		14
	Methanol		0	-	10	-	11		11
	Aqueous		-	-	-	-	-		12
*Argemone Mexicana* *leaf*	Ethyl acetate	*Citrobacter*	-	-	-	-	-		-
	Methanol		0	6	11	12	13		14
	Aqueous		0	-	10	11	13		14
*A. mexicana fruit*	Ethyl acetate		0	5	9	12	15		18
	Methanol		0	8	10	10	11		13
	Aqueous		-	-	-	-	-		11
*A. mexicana leaf*	Ethyl acetate	*Flexineri*	-	-	-	-	-	-	-
	Methanol		0	8	11	13	14		14
	Aqueous		-	-	9	10	14		14
*A. mexicana fruit*	Ethyl acetate		-	-	-	-	-		16
	Methanol		-		6	-	9	-	12
	Aqueous		-	-	-	-	-	-	-
*A. mexicana leaf*	Ethyl acetate	*S. epidermis*	0	-	-	9	13	-	16
	Methanol		-	-	-	-	-	-	16
	Aqueous		0	-	10	12	13		13
*A. mexicana fruit*	Ethyl acetate		-	-	6	11	12		15
	Methanol		-	-	-	-	-		13
	Aqueous		-	-	-	-	-	-	-

## Data Availability

Not applicable.

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
