# Peer review of "Analysis of Biochemical and Antimicrobial Properties of Bioactive Molecules of *Argemone mexicana"

_molecules, 2023, doi:10.3390/molecules28114428_

Round 1

Reviewer 1 Report

My main quastion is about how to use this medicine in patients? Can it be used as an ointment or pill or inhaler? In my opinion, this article needs to be pointed out and clarified about the mentioned topic.

Best regard

Author Response

 Reviewer-1

My main quastion is about how to use this medicine in patients? Can it be used as an ointment or pill or inhaler? In my opinion, this article needs to be pointed out and clarified about the mentioned topic.

It is used as an ointment for external use in the treatment of wound healing, the root paste is used to treat insect and scorpion bites etc. For constipation and bloating the root powder is used orally at the dose of 1-2 grams/day. The fruit extracts of A.mexicana have been used as intraperitoneal injection in anticancer studies in mice. (https://www.easyayurveda.com/2015/03/13/argemone-mexicana-uses-side-effects-research/#medicinal_properties).  This has been incorporated in the introduction and the reference has been included in the list of references.

Jain, R., Pandey, A., & Jain, R.S. (2011). EVALUATION OF ARGEMONE MEXICANA FRUITS EXTRACT USING MICRONUCLEUS ASSAY IN MOUSE BONE MARROW CELLS.Bulletin of Pharmaceutical Research 2011;1(2):22-4

Reviewer 2 Report

The text format is not the same in the all text. For example"Secondary metabolites of A. mexicana, including flavonoids, polyphenols, phenols, alkaloids, saponins, and tannins, are responsible for its medicinal properties" 

Author Response

Reviewer-2

The text format is not the same in the all text. For example "Secondary metabolites of A. mexicana, including flavonoids, polyphenols, phenols, alkaloids, saponins, and tannins, are responsible for its medicinal properties" .

The text format has been unified to A. mexicana.

Rev-3

the aim of the study should be more detailed, and novelty is questionable. Justify novelty in Introduction section

Discussion is not 'in depth’ and should incorporate more recent literature, the discussion needs to be rewritten in context of the prescribed aims of the manuscript.

The aim of this study was to optimize the best solvent and temperature for the extraction of secondary metabolites from different parts namely leaves, stem, fruits, and flowers, of A.mexicana and explore their antimicrobial and anti-HIV reverse transcriptase activities. This has been included in the introduction and the discussion has been elaborated.

Author Response

Reviewer-3

  1. The introduction has been elaborated and the aim of the study has been clarified.
  2. The UV-visible absorption spectroscopy results have been moved to supplementary figures section and chromatography data and figures have been included.
  3. The extraction was carried out for all the solvents at room temperature (identical for all solvents) and at their respective boiling points.
  4. To carry out the antibacterial study with individual constituents of the extract is beyond the scope of the present study and could not be carried out.
  5. In tables 5 and 6, the concentration of Ampicillin (5µg/mL) has been corrected.
  6. Discussion has been elaborated.
  7. The text has been formatted and the reference of Jaiswal et al., corrected to 2022.

Reviewer 4 Report

This manuscript demonstrates antioxidant, antiviral, and antimicrobial activities of A. mexicana plant extracts obtained using 4 different solvent systems at room or high temperature.  The results suggest usage of these extracts to identify active compounds in searches for potential candidates for future anti-infective agents.  These are questions that I would need authors to provide answers in order to continue to the next review process.

01.  There are many inconsistent formatting, punctuation, grammatical errors that authors need to ensure corrected.  For example, in Figure 1 “Leaf” and “Stem” panels, the Y-axis label should read “Absorbance”, not “Wavelength”.

02.  In page 03, please provide ATCC # for all bacterial strains for as it is a standard practice to provide these information.

03.  In page 03, since authors used well assays, authors should include technical replicates, not just biological replicates to improve statistical comparison.

04.  In page 03, authors should include an active plant extract as a positive control, as ampicillin is not a plant extract, so it is not a direct comparison for better or worse antimicrobial activity.

05.  In page 03, why do flavonoids and pheophytin A peaks show at different wavelengths among leaf, stem, and fruit extracts?  Should these peaks not show at consistent wavelengths as these compounds are solvated and out of localization effect (i.e., in leaf, stem, or fruit)?  Unless there are major differences in types of flavonoids and pheophytin A in different extracts, the average absorption spectra should be relatively similar among all extracts.  Please provide some insights on this.

06.  In page 04, are section 3.1 and 3.2 headings accidentally repeated or is this intended?  Please clarify.

07.  In page 11 (inconsistent page numbering, please fix it), antimicrobial activity seems not to be limited to preparation temperature and extraction medium, as no consistent trend was observed.  Please provide some insights on this.

08.  In page 15, please provide some insights, e.g., possible mechanism of action (MOA), and possible active compounds.

09.  Different types of antimicrobial activity assays can indicate possible different MOAs.  Inhibition zone assays only show which extract can actively inhibit bacterial growth or spread.  Bactericidal assays show which extract can actively reduce bacterial load after bacterial growth established.  I recommend authors to run another antimicrobial assay different from inhibition assay to get better comparison among extracts.

Please answer and correct the manuscript.  I would recommend continuing further review process as this manuscript may be of interests to some readers in anti-infective drug discovery fields.

Author Response

Reviewer 4

This manuscript demonstrates antioxidant, antiviral, and antimicrobial activities of A. mexicana plant extracts obtained using 4 different solvent systems at room or high temperature.  The results suggest usage of these extracts to identify active compounds in searches for potential candidates for future anti-infective agents.  These are questions that I would need authors to provide answers in order to continue to the next review process.

  1. There are many inconsistent formatting, punctuation, grammatical errors that authors need to ensure corrected.  For example, in Figure 1 “Leaf” and “Stem” panels, the Y-axis label should read “Absorbance”, not “Wavelength”.

Response- This has been corrected.

  1.  In page 03, please provide ATCC # for all bacterial strains for as it is a standard practice to provide this information.

 Response: Could not manage to get ATCC number

  1. In page 03, since authors used well assays, authors should include technical replicates, not just biological replicates to improve statistical comparison.

Response:This is beyond the scope now that the experiments have been completed.

  1.  In page 03, authors should include an active plant extract as a positive control, as ampicillin is not a plant extract, so it is not a direct comparison for better or worse antimicrobial activity. Response: Ampicillin being an antibiotic was used as a control because the pure and active extract/component was not available.
  2.  In page 03, why do flavonoids and pheophytin A peaks show at different wavelengths among leaf, stem, and fruit extracts?  Should these peaks not show at consistent wavelengths as these compounds are solvated and out of localization effect (i.e., in leaf, stem, or fruit)?  Unless there are major differences in types of flavonoids and pheophytin A in different extracts, the average absorption spectra should be relatively similar among all extracts.  Please provide some insights on this.

Response:In supplementary table 1, the range of absorbance of light by flavonoids and pheophytin A have been demonstrated by some workers as cited in last column of table. They have reported these molecules absorbing the light at different wavelengths.

  1.  In page 04, are section 3.1 and 3.2 headings accidentally repeated or is this intended?  Please clarify.

Response:This has been corrected.

  1.  In page 11 (inconsistent page numbering, please fix it), antimicrobial activity seems not to be limited to preparation temperature and extraction medium, as no consistent trend was observed.  Please provide some insights on this.

Response: The results obtained have been tabulated.

  1.  In page 15, please provide some insights, e.g., possible mechanism of action (MOA), and possible active compounds.

Response: The compound  3H-1,2,4-Triazole-3-thione, present in LCMS analysis of A. mexicana   leaf extracts  which is similar to  1,2,4-Triazole-3-thione is responsible for inhibion of  Staphylococcus aureus ATCC 6538, Staphylococcus epidermidis ATCC 12228, Escherichia coli ATCC 25922, Klebsiella
pneumoniae
ATCC 4352, Pseudomonas aeruginosa ATCC 27853, Proteus mirabilis
ATCC 14153 and Candida albicans ATCC 10231 reported in  (Çalişir et al., 2010) (M. M. Çalişir, B. Koçyiğit-Kaymakçioglu*, B. Özbek§ And G. Ötük, Synthesis and Antimicrobial Activity of Some Novel Schiff Bases Containing 1,2,4-Triazole-3-thione, E-Journal of Chemistry, 2010, 7(S1), S458-S464)

  1.  Different types of antimicrobial activity assays can indicate possible different MOAs.  Inhibition zone assays only show which extract can actively inhibit bacterial growth or spread.  Bactericidal assays show which extract can actively reduce bacterial load after bacterial growth established.  I recommend authors to run another antimicrobial assay different from inhibition assay to get better comparison among extracts.

Response - We agree with the comment of the honorable reviewer. However, the extracts of the different parts of the plants are finished and it may take more time to prepare the new extracts for the antimicrobial assay as suggested.

Please answer and correct the manuscript.  I would recommend continuing further review process as this manuscript may be of interests to some readers in anti-infective drug discovery fields.

Reviewer 5 Report

the aim of the study should be more detailed, and novelty is questionable. Justify novelty in Introduction section

Discussion is not 'in depth’ and should incorporate more recent literature, the discussion needs to be rewritten in context of the prescribed aims of the manuscript.

Author Response

Reviewer 5

the aim of the study should be more detailed, and novelty is questionable. Justify novelty in Introduction section

Response:  Significant HIV potential seen in the extract that can be used in developing therapeutics

Discussion is not 'in depth’ and should incorporate more recent literature, the discussion needs to be rewritten in context of the prescribed aims of the manuscript.

Newly added matter is indicated in red.

Round 2

Reviewer 3 Report

I think that the authors answered the questions formally and when revised  the article took into account the less significant comments of the reviewers. The main remark “The introduction to the article should show the relevance of the study and the current level of the problem being solved. In the present work, this section is written very sparingly and is based on outdated literature data” was left without attention. I will add that a similar remark was made by the fifth reviewer. 

Author Response

As per the suggestion of the reviewer the introduction and the aim of the study has been elaborated which is indicated in red in the revised manuscript.  I am attaching the revised with this email also   Thanking you Best regards   Nikhat J Siddiqi

Reviewer 5 Report

accept

Author Response

-